# Femtosecond photoexcitation dynamics inside a quantum solvent

Bernhard Thaler[1], Sascha Ranftl[1], Pascal Heim[1], Stefan Cesnik[1], Leonhard Treiber[1], Ralf Meyer [1], Andreas W. Hauser [1], Wolfgang E. Ernst [1] & Markus Koch [1]

The observation of chemical reactions on the time scale of the motion of electrons and nuclei has been made possible by lasers with ever shortened pulse lengths. Superfluid helium represents a special solvent that permits the synthesis of novel classes of molecules that have eluded dynamical studies so far. However, photoexcitation inside this quantum solvent triggers a pronounced response of the solvation shell, which is not well understood. Here, we present a mechanistic description of the solvent response to photoexcitation of indium (In) dopant atoms inside helium nanodroplets ($He_N$), obtained from femtosecond pump–probe spectroscopy and time-dependent density functional theory simulations. For the In–$He_N$ system, part of the excited state electronic energy leads to expansion of the solvation shell within 600 fs, initiating a collective shell oscillation with a period of about 30 ps. These coupled electronic and nuclear dynamics will be superimposed on intrinsic photoinduced processes of molecular systems inside helium droplets.

---

[1] Institute of Experimental Physics, Graz University of Technology, Petersgasse 16, 8010 Graz, Austria. Correspondence and requests for materials should be addressed to M.K. (email: markus.koch@tugraz.at)

Since the award of the 1999 Nobel Prize for Chemistry[1], various fundamental molecular processes have been investigated on their natural time scales, e.g., fragmentation via different pathways on the molecular potential energy surface[2], non-adiabatic electron-nuclear coupling[3], or electron dynamics initiated by ultrashort laser pulses[4]. Superfluid helium nanodroplets ($He_N$) have been used as nanocryostats to isolate atoms or molecules at 0.4 K temperature, or to form new weakly bound aggregates[5,6]. Their gentle influence on guest particles is demonstrated, for example, by electron spin resonance[7] or molecular rotation and alignment experiments[8,9]. He droplets are an appealing spectroscopic tool because of their transparency for electromagnetic radiation up to the extreme ultraviolet energy regime[5]. However, photoexcitation inside the droplet leads to dissipation of significant excess energy via coupling to collective modes of the surrounding helium, which is expected to be a fast process. Femtochemistry inside $He_N$ will allow real-time tracking of photochemical reactions in novel systems, such as fragile agglomerates[10–12], or molecules in a microsolvation environment[13]. This will, however, require a detailed knowledge about the response of the quantum fluid to the photoexcitation of a dopant atom or molecule. So far, only the ultrafast dynamics in pure helium droplets have been studied[14], and femtosecond measurements on doped helium droplets were restricted to the surface-bound alkali metals[15,16] that can hardly couple to helium bubble modes. Since most foreign atoms and molecules reside inside the droplets and couple more strongly, we have concentrated on the electronic excitation of single atoms well inside the droplets. In this way, no other degrees of freedom such as rotation or vibration would interact and only the coupling of the electronic excitation with the modes of the surrounding helium should be detected. Previous spectroscopic studies in the frequency domain have shown blue-shifted excitation bands of dopants inside droplets compared to gas phase indicating that an excess energy is required to create a correspondingly larger helium bubble to accommodate the excited electron orbital[5]. This excess energy must be released to the helium in the form of a damped helium excitation mode.

In our work, we follow the expansion of the helium bubble after electronic excitation of single indium (In) dopants in real time. After an expansion from 4.5 to 8.0 Å radius in 600 fs, we observe a contraction of the surrounding He at (28 ± 1) ps, as well

as an ejection of the dopant atom from the droplet about 60 ps after the electronic excitation. As observable in our femtosecond pump–probe measurements, we chose the photoelectrons released because they have been shown to exit the droplet rather ballistically without being significantly influenced by the helium environment[14,15,17]. In spite of its importance for photochemical studies in superfluid helium droplets, this sequence of events has not previously been observed.

## Results

**Photoexcitation dynamics of the In-$He_N$ system.** We investigate photoexcitation dynamics of the In–$He_N$ system with a combination of time-resolved photoelectron spectroscopy (TRPES) and time-dependent density functional theory (TDDFT) simulation, as described in the following. A mechanistic description of the processes deduced from experiment and theory will be discussed in the final paragraph.

**Time-resolved photoelectron spectroscopy.** The feasibility of ultrafast experiments inside $He_N$ ultimately depends on the availability of an experimental observable that is available with sufficiently low distortion by the intermediate helium. Ion detection, as used on the droplet surface, is not possible because ions are captured inside the droplet due to their attractive potential[15]. Photoelectron (PE) detection, in contrast, has been successfully used for pure and doped $He_N$[14,15,17]. TRPES is a well-established method for ultrafast gas-phase studies and is primarily sensitive to the electronic structure of a system[18,19]. As depicted in Fig. 1a, after photoexcitation by a pump pulse, the evolution of the excited state is probed by time-delayed photoionization and the PE kinetic energy (red arrows) is measured. When applied inside a $He_N$, photoexcitation induces an abrupt disturbance of the quantum fluid solvation shell due to the expansion of the valence electron wave function. Because the energies of the electronic states depend on the structure of the He environment, the transient response of the quantum solvent can be sensed with TRPES (see Fig. 1a).

Figure 2 shows the time-dependent evolution of the PE signal within the first picosecond after photoexcitation (a), together with PE spectra at selected pump–probe times (b). Within about 600 fs, the PE peak energy is shifted from 0.61 to 0.34 eV, followed by

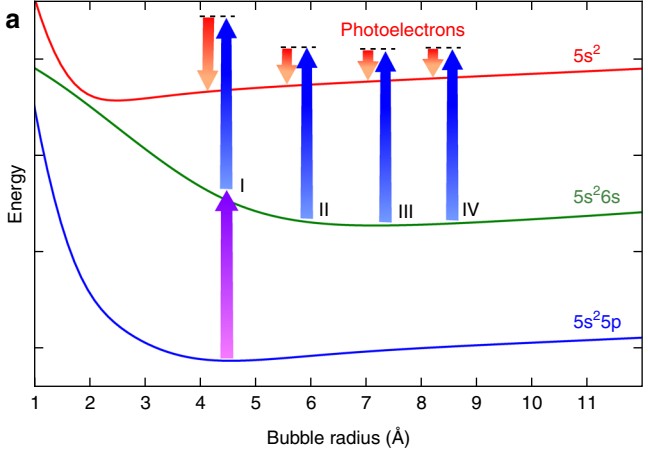
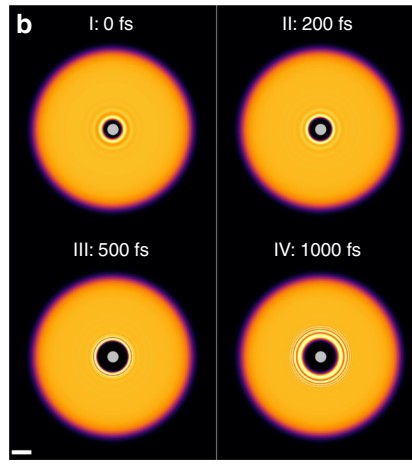

**Fig. 1** Temporal evolution of the In–$He_N$ system after photoexcitation. **a** Sketch of the In–$He_N$ potential energy surfaces as a function of the bubble radius for In in its ground [$5s^25p$ ($^2P_{1/2}$), blue], lowest excited [$5s^26s$ ($^2S_{1/2}$), green], and ionic ground state [$5s^2$ ($^1S_0$), red]. The purple arrow indicates pump excitation at 376 nm, blue arrows indicate probe ionization at 405 nm for characteristic delay times, and red arrows correspond to the PE kinetic energy, as measured by TRPES. **b** Helium density distributions of a $He_{4000}$ droplet with an In atom located in the center for selected times after photoexcitation, as calculated with TDDFT. Scale bars, 10 Å

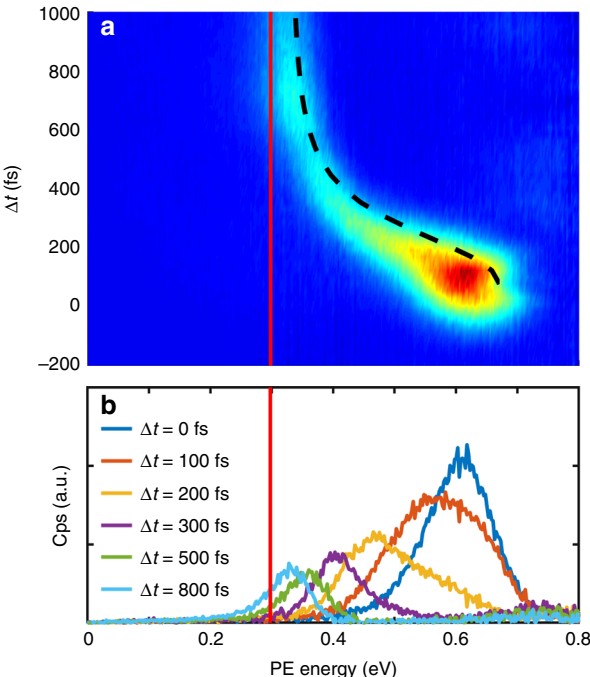

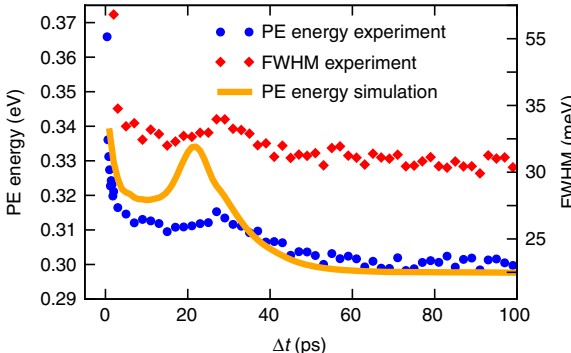

**Fig. 2** Time-resolved photoelectron (PE) spectra of single In atoms solvated inside $He_N$. The average droplet size is 4000 He atoms. **a** PE kinetic energy spectrum as the function of the pump–probe time delay $\Delta t$, together with the simulated dynamics (dashed line) and the gas-phase PE energy (solid line). Around time-zero the PE signal is increased due to temporal overlap of the pump and probe pulses. Additionally, the total PE signal decreases during the expansion, which might be due to a decreased ionization probability for larger bubbles and/or lower escape probability of slow electrons from larger bubbles at long delays compared to fast electrons from small bubbles at short delays[17, 31]. **b** Selected spectra for different pump–probe time delays, which resemble horizontal cuts through the 2D plot in **a**

**Fig. 3** Photoelectron (PE) peak position and linewidth as a function of time delay $\Delta t$. The transient peak position (blue dots) and full width at half maximum (FWHM, red diamonds) are shown within 100 ps after photoexcitation, as measured with TRPES and simulated with TDDFT (orange line). The experimental peak position and FWHM are obtained by Gaussian fits to the corresponding PE energy spectra. The start position for the TDDFT simulation was 20 Å from the droplet center in order to obtain a similar ejection behavior as the experiment

a slower decrease to 0.32 eV at 1000 fs, which is about 0.02 eV above the gas-phase peak that appears at around 0.30 eV (solid line in Fig. 2). The remaining shift represents the reduced ionization potential of In atoms in the He environment due to polarization effects[17]. The linewidth of the PE spectra is significantly increased and changes within the first picosecond (Fig. 2b), which we ascribe to the following four reasons: First, during pump–probe cross-correlation of 150 fs, saturation effects and the spectral width of the pump pulse are expected to contribute to the PE linewidth. Second, within the first 500 fs, a peak shift with a maximum slope of about 1 meV/fs in combination with the 150 fs pump–probe cross-correlation leads to an expected contribution of about 150 meV. Third, ionization inside the droplet increases the linewidth, given by the Franck–Condon overlap of the excited and the ionic state (c.f., Fig. 1), which seems to be the dominant contribution to the linewidth after 500 fs. Fourth, relaxation of the photoelectrons due to binary collisions with individual He atoms on the way out of the droplet leads to an asymmetric shape of the PE peaks[17]. These decelerated electrons can be seen as wing extending to PE energies below the gas phase value (red line in Fig. 2b and Supplementary Fig. 7b).

In Fig. 3, the PE kinetic energy up to 100 ps is shown (blue dots). After a steep decrease representing the tail of the initial peak shift shown in Fig. 2, the peak position slowly decreases to reach a constant value at about 60 ps with a temporary increase at $(28 \pm 1)$ ps. The PE peak width shows a very similar trend (Fig. 3,

red diamonds) with a steady decrease over time to about 35 meV at long time delays and a temporary increase. Detailed scans of PE peaks at short and long time delays are shown in Supplementary Fig. 7b. We note that except for very short time delays right after the pump–probe overlap (cross-correlation), the total PE yield stays constant over the whole investigated temporal region.

**Time-dependent helium density functional theory**. To obtain further insight into the ultrafast dynamics, photoexcitation of the In–$He_N$ system is simulated with TDDFT using the BCN-TLS-He-DFT computing package[20], which has been successfully applied to reproduce the dynamics of $He_N$ loaded with various different atomic species[21]. In the present case, an extraordinary amount of excess energy of several hundred meV is coupled into the system in the photoexcitation process. We therefore carefully tested the simulations for convergence by variation of the simulation parameters (see Supplementary Note 3 and Supplementary Figs. 4, 5).

Figure 1b shows He density distributions for selected times after photoexcitation and the corresponding bubble expansion over time is plotted in Fig. 4a. Inside the droplet, the energies of the In excited state ($5s^26s$) and its ionic state ($5s^2$) deviate from the bare atom values by the interaction energies $E_{He_N-In^*}$ and $E_{He_N-In^+}$, respectively. These interaction energies, plotted in Fig. 4b, are calculated by integrating the respective In–He pair potentials over the He density. While $E_{He_N-In^*}$ (cyan curve) is positive and decreases with time (for larger bubbles), $E_{He_N-In^+}$ (red curve) is negative and increases. This behavior can be expected from the repulsive and attractive character of the excited and the ionic state pair potentials, respectively (Supplementary Fig. 3). The simulated PE peak shift with respect to the free atom, as plotted in Fig. 4c, is calculated as the difference of the two interaction energies ($E_{He_N-In^*} - E_{He_N-In^+}$) and compared to the measured transient peak shift in Figs. 2a, 4c, revealing good agreement. Note that within 1000 fs $E_{He_N-In^*}$ decreases to zero, whereas $E_{He_N-In^+}$ is negative and reaches zero only at higher time delays. This results in a further peak shift between 1000 fs and 60 ps (see Fig. 3), as the dopant is ejected from the droplet. As can be seen in Fig. 4c, below 200 fs the experimental peak shifts are slightly lower than the simulated ones, which we ascribe to a distortion of the PE peaks due to a cross-correlation signal caused by the overlap of pump and probe pulses in this temporal region (c.f., Fig. 2b).

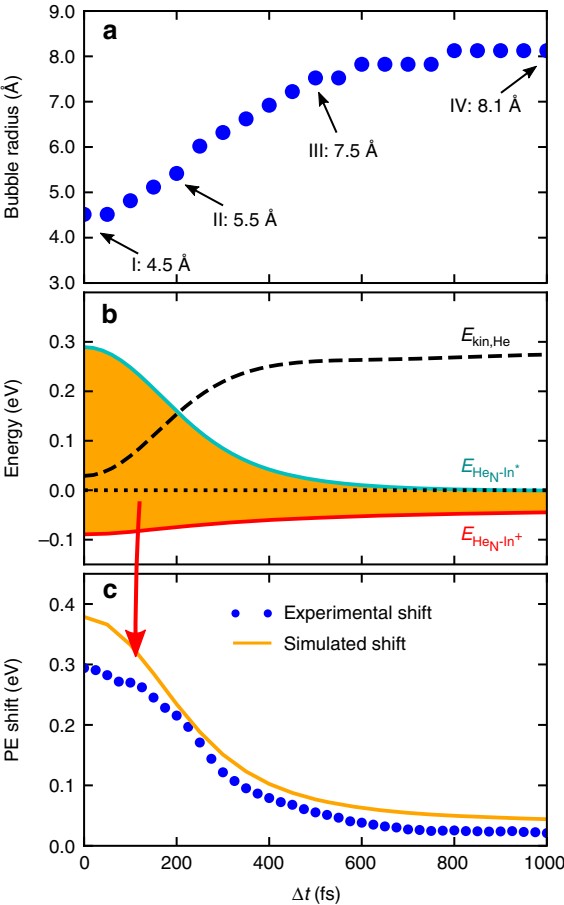

**Fig. 4** Photoexcitation (PE) dynamics of the In–He$_{4000}$ system simulated with TDDFT. **a** Bubble radius as a function of time delay $\Delta t$, determined as the position of the corresponding He distribution at which the density has dropped to 50% of the bulk value. Times for which the calculated He density is shown in Fig. 1b are indicated. **b** Interaction energy $E_{\text{He}_N-\text{In}^*}$ of the $5s^2 6s$ excited state (cyan curve) and interaction energy $E_{\text{He}_N-\text{In}^+}$ of the $5s^2$ ionic state (red curve). Additionally, the kinetic energy of the He atoms, $E_{\text{kin,He}}$, is plotted as dashed line. **c** Simulated PE peak shift induced by the He environment (orange line), obtained as $E_{\text{He}_N-\text{In}^*} - E_{\text{He}_N-\text{In}^+}$ (indicated by the shaded area in **b**), which is also shown in Fig. 2a. For comparison to the measured shift of the PE peak position over time, the recorded electron spectra at all time delays (c.f., Fig. 2b) are fitted with Gaussian functions and the line positions are indicated here by blue dots

Next, we compare the steady decrease of the excited state electronic energy (cyan curve in Fig. 4b) to the kinetic energy of the helium atoms (dashed line in Fig. 4b), and find that the two curves show almost exactly complementary trends.

Finally, the simulated PE peak position for an In atom, that is photoexcited at a distance of 20 Å from the droplet center, is shown in Fig. 3 (orange line). The choice of this position is justified by comparing simulated PE peak transients with different starting positions (see Supplementary Note 4 and Supplementary Fig. 6). The simulated curve shows the same overall decrease as the experimental values (blue dots), although with a more pronounced temporal increase at 22 ps.

## Discussion

The transient shift in the pump–probe PE spectrum of the In–He$_N$ system within the first picosecond (Fig. 2) has to be related to solvation shell dynamics, as no internal degrees of freedom are available for relaxation of the In atom in its lowest electronically excited state. The energy of the excited valence electron in the In*–He$_N$ system is a very sensitive probe for the temporal evolution of the He environment because of strong Pauli repulsion with the surrounding helium[22]. TRPES measures the transient PE kinetic energy, which additionally depends on a temporal shift of the ionic state energy ($E_{\text{He}_N-\text{In}^+}$, Fig. 4b). Therefore, we use TDDFT modeling of the photoexcitation process in order to distinguish these two contributions. Previously, TDDFT simulations could only be compared to time-dependent experiments at the weakly-interacting droplet surface[16]. In the interior, the dopant-He interaction is much stronger, with the consequence that significantly more excess energy (270 meV ≈ 2200 cm$^{-1}$ in our case) is coupled into the system during photoexcitation, challenging the accuracy of the TDDFT approach. The reproduction of the observed transient PE peak shift by TDDFT (Figs. 2a, 4c), without using any experimental input for the simulation, demonstrates that a simulation of photoexcitation dynamics is possible even in the case of significant excess energy.

By combining experiment and theory, we obtain the following mechanistic picture of the coupled, ultrafast electronic and nuclear relaxation process: Photoexcitation increases the radial expansion of the valence electron wave function, as is suggested by the strong repulsive part of the In–He pair potential at short distances in the excited state (Supplementary Fig. 3). Pauli repulsion between the extended electron density and the closed-shell He thereby pushes the surrounding solvation shell away (see Supplementary Movie 1). The spherical He bubble containing the excited In atom almost doubles its radius from 4.5 to 8.1 Å within 600 fs after excitation (Figs. 1b, 4a). This process can also be explained with the corresponding potential energy surfaces (Fig. 1a): Because the equilibrium bubble radius of the excited electronic state is larger than that of the ground state, photoexcitation causes enlargement of the solvation shell. This nuclear relaxation can be followed as transient PE peak shift because the potential energies of the excited state and the ionic state depend on the distance of neighboring He atoms to the In dopant. From an energetic viewpoint, the bubble expansion is accompanied by the conversion of electronic energy into kinetic energy of the He atoms, as illustrated by the mirror-imaged progression of the two corresponding curves (excited state interaction energy $E_{\text{He}_N-\text{In}^*}$ and kinetic energy of the He atoms $E_{\text{kin,He}}$) in Fig. 4b. The minute decrease of the sum of $E_{\text{He}_N-\text{In}^*}$ and $E_{\text{kin,He}}$ over time represents energy transferred to He–He interactions (correlation energies).

The impulsive stimulation of the He solvation layer initiates a collective oscillation of the He bubble, the first contraction of which is observed as an increase of the PE kinetic energy and linewidth in Fig. 3 at (28 ± 1) ps, induced by the temporally increased He density in the vicinity of the In atom. The repulsive character of the excited state In–He pair potential (see Supplementary Note 2 and Supplementary Fig. 3) leads to ejection of the In atom from the droplet on a time scale of about 60 ps (see Supplementary Movie 2). Consequently, the PE kinetic energy decreases to the free-atom value within this time span (see Fig. 3) and only one bubble oscillation can be observed. Dopant ejection is further confirmed by observing a rise in photoion yield on the same timescale (Supplementary Fig. 7a) and a transient change of the linewidth of the PE peak (see Fig. 3 and Supplementary Fig. 7b). While the TDDFT simulation assumes a fixed starting location of the In atom, the experimentally observed ensemble comprises a distribution of In atoms within the droplet. As a consequence, the timing of the first bubble contraction will appear smeared out in the experimental data, because the PE energy peak shift due to dopant ejection is superimposed on the pure bubble oscillation. Photoexcitation of the In dopant in the center of the droplet induces multiple oscillations and no ejection

within the simulated time span (see Supplementary Movie 3 and Supplementary Note 4). We therefore conclude, that the collective solvation shell oscillation has a period of about 30 ps, the observation of which provides insight into the hydrodynamics of the bubble in real time[23].

In conclusion, our experiments prove that ultrafast, coupled electronic and nuclear dynamics of particles located inside superfluid He nanodroplets can be observed and simulated. The expansion of the dopant solvation shell will be superimposed on any molecular relaxation dynamics on femtosecond time scales inside the droplet. When applying photoelectron detection, which seems to be a promising observable for intrinsic molecular dynamics inside helium droplets, the photoelectron transients induced by solvation shell dynamics have to be known. The duration of dopant ejection, on the other hand, limits the time frame for which ultrafast reactions inside the quantum fluid can be observed. As a proof of concept, our results pave the way to use helium droplets as a novel sample preparation technique for ultrafast studies on previously inaccessible tailor-made or fragile molecular systems.

## Methods

**Helium droplet generation and In atom pickup**. Helium droplets with an average size of about 4000 atoms are generated by supersonic expansion of high purity (99.9999%) helium gas through a cooled nozzle (5 μm diameter, 18 K temperature, 40 bar stagnation pressure) into high vacuum. The expansion in combination with evaporative cooling results in droplet temperatures of about 0.4 K, which is well below the superfluid phase transition of helium. The $He_N$ are doped with In atoms inside a pickup region, where indium is resistively heated. Pickup conditions are optimized for single atom pickup and for an acceptable signal-to-noise ratio. Indium was chosen as dopant because of its simple electronic structure with one valence electron and because its excited state is symmetric, simplifying the TDDFT simulations, as well as the interpretation. After passing a differential pumping stage to increase the vacuum quality, the doped droplets enter the measurement chamber, where the $He_N$ beam is crossed at a right angle by the femtosecond laser pulses inside the extraction region of a time-of-flight spectrometer.

**Time-resolved photoelectron spectroscopy**. A commercial Ti:sapphire femtosecond laser system (Coherent Vitara oscillator, Legend Elite Duo amplifier) delivers 25 fs laser pulses with 800 nm central wavelength and 4 mJ pulse energy at a repetition rate of 3 kHz. The pulses are split into a pump and a probe path with variable time delay. Pump pulses are frequency up-converted by an optical parametric amplifier (Coherent OPerA Solo) that tunes the wavelength to 376 nm (3.30 eV, 6 nm ≈ 60 meV full width at half maximum, FWHM). Probe pulses are frequency doubled to 405 nm (3.06 eV) with a 1 mm thick BBO crystal (3 nm ≈ 25 meV, FWHM) for short delays and with a 5 mm thick LBO crystal (1.5 nm ≈ 10 meV, FWHM) for long delays and guided over a delay stage. Dichroic mirrors are used in both beam paths to remove undesired wavelengths from the upconversion process. Pump and probe pulses are focused into the extraction region of the linear time-of-flight spectrometer, where they overlap in space and time at the intersection region with the $He_N$ beam. A magnetic bottle configuration[24] ensures high electron detection efficiency and a small positive repeller voltage of a few hundred mV increases the electron kinetic energy resolution. At these parameters, we estimate the relative energy resolution of the spectrometer to be about 10%, based on reference measurements. PE energies are calibrated with the free atom line, which position is retrieved by subtraction of the excited state binding energy[25] from the probe photon energy. The measurement chamber is operated at a base pressure of $10^{-10}$ mbar.

The intensities of the pump and probe pulses are optimized to obtain a maximum pump–probe signal with respect to pump-only and probe-only backgrounds. The pump wavelength for In excitation to the lowest excited state ($5s^2 6s$) is chosen to be 376 nm in order to optimize the monomer to dimer ratio (see Supplementary Note 1 and Supplementary Fig. 2), which is blue-shifted by 270 meV with respect to the gas-phase excitation wavelength at 410 nm[25]. This amount of excess energy is coupled into the In–$He_N$ system at photoexcitation. The pump–probe cross-correlation is estimated with 150 fs.

**Time-dependent helium density functional theory**. In the last years, the approach of TDDFT for the bosonic system of helium has been successfully applied to describe the dynamical interaction of surface- and center-located species with the helium quantum fluid, providing important insight into effects like superfluidity on the microscopic level[26], desorption dynamics[16], or collision processes[27,28].

Details on the application and formalism of static and dynamic HeDFT are given elsewhere[21] and the computing package of the BCN-TLS group is available to the public as open source[20]. Here, only the basic concepts and the terms that affect the presented results are given: Both static and dynamic computations are based on the Orsay–Trento functional[29], which attributes for He–He interactions, and the diatomic In–He potential energy surfaces. These pair potentials were calculated with high-level ab initio methods for the ground, excited, and ionic state (see Supplementary Note 2 and Supplementary Fig. 3). The simulations are performed for a $He_{4000}$ droplet with the In impurity located in the center by using a He-functional that includes the solid term[30]. We use a three-dimensional Cartesian box of 96 Å length with a discrete grid size of 320 pt (0.3 Å spacing) and time steps of 0.1 fs to simulate the bubble expansion dynamics within the first picosecond and a grid size of 256 pt (0.375 Å spacing) and time steps of 1 fs for the bubble oscillation dynamics up to 100 ps. For the bubble oscillation dynamics, the starting position was chosen to be at 20 Å distance to the center, which leads to a similar ejection behavior as in the experiment. Both the bubble expansion and the oscillation period are local effects and are found to be very similar for dopant locations in the droplet center. With the statically optimized ground state He density, a dynamical evolution is triggered by replacing the ground state pair potential with the excited state pair potential. This instantaneous perturbation drives the system and TDDFT allows to follow the resulting dynamics in real time[21], by solving the TDDFT equations for the helium and Newton's equations of motion for the impurity. Photoelectron spectra are simulated by integrating the pair potential energies $E_{He–In}$ over the whole droplet density $\rho_{He}$ for both the excited and the ionic state for various time steps in the simulation. The difference between the interaction energies directly compares to the difference in ionization energy of the immersed impurity and therefore to the shift in PE energy:

$$PE \, shift(t) = \int \rho_{He}(\mathbf{r}, t) E_{In^*-He}(\mathbf{r} - \mathbf{r}_{In^*}(t)) d\mathbf{r}$$
$$- \int \rho_{He}(\mathbf{r}, t) E_{In^+-He}(\mathbf{r} - \mathbf{r}_{In^+}(t)) d\mathbf{r}$$

Since a huge amount of energy is deposited into the system in the excitation process, the simulations were tested for numerical uncertainties by variation of different parameters (grid size, time step, and cutoff energy), as presented in Supplementary Note 3 and Supplementary Figs. 4–5.

## Data availability

The data measured, simulated, and analyzed in this study are available from the corresponding author on reasonable request.

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

## Acknowledgements

We thank Manuel Barranco, Martí Pi, and the whole BCN-TLS-HeDFT-code team for their support with technical details of the simulations and fruitful discussions. We appreciate the experimental support by Miriam Meyer and acknowledge financial support by the Austrian Science Fund (FWF) under Grant P29369-N36, as well as support from NAWI Graz.

## Author contributions

M.K. conceived and designed the experiment; P.H. built the experimental setup with contributions of B.T., S.R., and M.K.; B.T., S.R., S.C., and L.T. performed the experiment; B.T. performed the TDDFT simulations; R.M. and A.W.H. calculated the In–He pair potentials; W.E.E. provided the laser system; B.T. and P.H. analyzed the data; all authors contributed to the interpretation of the results; B.T., W.E.E., and M.K. wrote the paper with contributions of P.H.

## Additional information

**Competing interests:** The authors declare no competing interests.

