## [Peer Review File · Nature Communications]

Response Reviewer 1

1) *In the abstract one gets the impression that the oscillation period of 28 ps is a property of helium droplets. This is certainly not the case it is very specific for the system studied, i.e. exact location of the atom and the size of the droplets, and the authors should phrase this more carefully or even remove the reported value from the abstract.*

We agree that the observed time scales are exclusively for the indium–helium-droplet system and account for this by a more clear distinction between the observation of a solvation shell contraction and the intrinsic shell oscillation period throughout the whole paper. Because the shell oscillation is superimposed on dopant ejection, the observed contraction is slightly shifted in time with respect to the real oscillation period.

We therefore made the following changes:

- Abstract: "For the In-He_N system, part of the excited state electronic energy leads to expansion of the solvation shell within 600 fs, initiating a collective shell oscillation with a period of about 30 ps."
- Page 2: "After an expansion from 4.5 Å to 8.0 Å radius in 600 fs, we observe a contraction of the surrounding He at (28 ± 1) ps, as well as an ejection of the dopant atom from the droplet about 60 ps after the electronic excitation."
- Discussion (page 9): "As a consequence, the timing of the first bubble contraction will appear smeared out in the experimental data, because the PE energy peak shift due to dopant ejection is superimposed on the pure bubble oscillation. We therefore conclude, that the collective solvation shell oscillation has a period of about 30 ps, the observation of which provides insight into the hydrodynamics of the bubble in real time [23]."

Concerning the dopant location, we now mention in the paper that the oscillation period does not strongly depend on this parameter (see comment to point 10 below).

2) *On the bottom of page 2, it is stated that the bubble size oscillates with a period of 28 ps, this is different from what is stated in the abstract where it is attributed to a collective oscillation of the helium. Inspection of Movie 2 in the supplementary information to me indicates that the compression of the solvation shell after 28 ps is related to the excited atom being accelerated towards the surface and not to an oscillation of the droplet, let alone the solvation shell, with a 28 ps period.*

We have changed the wording of "collective oscillation of the helium" in the abstract. The simulations clearly show that the PE energy rise around 28 ps is solely attributed to the bubble contraction phase of the solvation shell oscillation. However, the dopant ejection is superimposed to that dynamics.

Setting the dopant to a starting location deeper inside the droplet reveals a very similar solvation shell oscillation and also leads to more than one oscillation period. As a detailed position dependence would go beyond the scope of this manuscript, we plan to address this interesting question in a follow-up publication. In this paper, we added four snapshots of supplementary video 2 (20 Å starting position) to the supporting online material, where one can clearly see a complete contraction of the solvation shell at 22 ps (the local maximum of the simulated peak shift, see fig. 3 in the paper), compared to the snapshot at 10 ps.

3) Figure 1a is missing values on the vertical axis, these should be added. In this same figure the energy for the 6s state does not seem to increase at large bubble radius. This is surprising because the formation of a large bubble requires quite some energy. How can this be?

Figure 1a shows a sketch of the potential energy surfaces (as stated in the caption) and not calculated potentials. We therefore cannot give energy values. We have removed the term "(a.u.)" from the vertical axis label, to avoid the impression that the curves represent real energies. We agree, however, that the state energies should increase for larger bubble radii, which has been corrected in figure 1.

4) On the top of page 5 it is stated that the photoelectron energy levels of at 0.32 eV. However, in Figure 3 the energy levels of at 0.31 eV. How can this be?

We agree that the wording "level out", describing figure 2, is misleading because the PE peak shift continues after 1 ps (the longest time delay shown in fig. 3). We have changed

"Within about 600 fs the PE peak energy is shifted from 0.61 eV to 0.34 eV. Afterwards the PE energy levels out around 0.32 eV."

to

"Within about 600 fs the PE peak energy is shifted from 0.61 eV to 0.34 eV, followed by a slower decrease to 0.32 eV at 1000 fs,"

5) The data points reported in Figure 3 have been extracted from the spectra by fitting the peaks to Gaussians. Inspection of the spectra in Figure 2 reveals that these not all have a shape compatible with a Gaussian distribution. How does this effect the data in figure 3?

We have checked this influence by determining the position as "center of gravity" (mean value) of the PE peak. Comparison of the "Gaussian" PE peak transient to the "center of gravity" PE peak transient gives a slight, but negligible, difference only within a short time period after the cross correlation, as there the PE spectra are a mixture of cross correlation and shifted peaks, representing the bubble expansion. After about 400 fs the PE peaks are represented very well by Gaussian functions. The data in figure 3, which focus on time scales far above 1 ps, are therefore not influenced by the fit.

6) The linewidth of the PE spectra is considerably broader than the instrumental width of a few percent; the authors do not provide an explanation for this fact. Related to this, how can it be that PE extends to energies below that of the gas phase. According to the model provided in Figure 4 this implies that the ionization energy, $E_{\text{Hen-In+}}$, becomes larger than the gas phase value, or the excited state energy $E_{\text{Hen-In}^*}$ smaller than the gas phase value. Both options seem unlikely.

The energy resolution of the PE spectra is limited by the bandwidth of the probe pulse (25 meV) and the energy resolution of our time-of-flight spectrometer (10 %, e.g., $300 \text{ meV} \cdot 0.1 = 30 \text{ meV}$). In combination we thus estimate $\sqrt{(25 \text{ meV})^2 + (30 \text{ meV})^2} = 39 \text{ meV}$ FWHM of the PE peaks, which is in agreement with the observed spectra in figure 2. Because of this finite line width, the PE signal can extend below the line indicating the gas phase position.

We have added the sentence "At these parameters we estimate the relative energy resolution of the spectrometer to be about 10%, based on reference measurements." to the methods section and

also added more details, including the bandwidth in meV, to the pulse parameters.

7) Inspection of figure 2 reveals that both the linewidth and intensity varies with time delay. The authors attribute this to a cross correlation signal at short delay times, i.e. below 200 fs. However, based on the reported bandwidth of the laser pulses a cross correlation of less than 100 fs is expected. Moreover, even at a delay time of 300 fs a significant broadening and reduction of electron signal is observed, questioning this interpretation of the data.

We agree that the PE signal drops in magnitude also after 200 fs, where only a minor influence of the cross correlation is expected. This indicates that the ionization probability depends on the bubble size, with a reduction for larger bubbles. This however remains a speculation at the moment and will be addressed in future studies. We have added the sentence "Additionally, the total PE signal decreases during the expansion, hinting at reduced ionization probabilities for larger bubbles." to the caption of figure 2. Concerning the increased linewidth, we attribute the broadening at e.g. 300 fs to a convolution of the molecular dynamics with the laser pulse duration (because of the pulse duration, not all bubble expansion "trajectories" of the observed ensemble are started at the same time).

The interpretation of the presented results is, however, not affected as it is based on the PE peak position, which is correctly determined.

8) In figure 3 the results are reported as experimental and simulated shift. As far as I understand the data, this is not a shift but the actual value of the photoelectron energy. The authors should use other symbols instead of crosses to make the figure better readable.

We agree, the data represent PE energies and not peak shifts. We have changed the figure legend. We further changed the markertype from crosses to dots in figures 3 and 4.

9) The authors attribute the decrease of the PE energy after 30 ps shown in Figure 3 to the excited atom leaving the droplet. In this case, the width of the PE peaks should become considerably narrower at long delay times as free atoms are formed. The authors should provide this information. At the same time they should provide details on the integrated signal levels, to rule out other possible processes taking place, see also my concerns above.

We indeed observe a slightly increased PE line width for photoionization inside the droplet (about 40 meV), as compared to the bare atom line (35 meV). For the interpretation of the dynamics, we chose to evaluate the PE line position because it is a sensitive measure for the He density in the vicinity of the dopant, and is also directly comparable to the TDDFT simulations. The position dependence of the line width, in contrast, cannot be easily modeled, for which reason we prefer not to discuss it in the paper.

Regarding the integrated PE signal, which we have measured, we added the sentence "We note that except for very short time delays right after the pump-probe overlap (cross-correlation), the total PE yield stays constant over the whole investigated temporal region." to the results-section.

10) Comparison of the TDDFT calculations and the experimental data lead to the conclusion that the increase of the PE energy at 28 ps is due to compression of the bubble structure surrounding the atom. The calculations largely reproduce this time scale assuming the atom is displaced 20 Å from the droplet center. It would be interesting to learn how this time shifts as a function of

position, especially since the authors claim that the effect is reduced in the experiment due to wide distribution of positions in the droplet. The authors should discuss the position distribution of the atom in the context of the work of Lehmann on this subject.

As is stated in the text, the 20 Å displacement is justified by a good agreement with the experiment, as deeper dopant locations give rise to much longer, or even absent ejection dynamics. Also, 20 Å represents the maximum of the spatial probability distribution in the ground state holding-potential. To verify that the solvation shell oscillation is comparable throughout the whole droplet interior, we carried out simulations at the droplet centre, where we find a very similar temporal behavior. However, a detailed investigation of the dependences of the dynamics on position and droplet size are beyond the scope of this manuscript and will be addressed in future studies.

We have extended our explanation in the methods-section to "For the bubble oscillation dynamics the starting position was chosen to be at 20 Å distance to the centre, which is, based on in the In-He_N ground state holding potential, a representative dopant location, and leads to a similar ejection behavior as in the experiment. Both the bubble expansion and the oscillation period are local effects and are found to be very similar for dopant locations in the droplet centre."

11) In Figure 4 the energy of the excited atom, $E_{\text{HeN-In}^}$, is plotted as time with respect to the free atom. In less than 1 ps this energy difference becomes zero. How is this compatible with figure 3 where after 60 ps one has free atoms, and the PE energy is 0.01 eV lower than at 1 ps?*

It is true that the excited energy reaches zero for higher bubble radii, however the ionic energy $E_{\text{HeN-In}^+}$ is still lower than for the gas phase value (due to helium-induced polarization effects), which causes the additional shift for higher time delays.

To emphasize this, we added the sentence "Note that within 1000 fs $E_{\text{HeN-In}^*}$ decreases to zero, whereas $E_{\text{HeN-In}^+}$ is negative and reaches zero only at higher time delays. This results in a further peak shift between 1000 fs and 60 ps (see figure 3), as the dopant is ejected from the droplet." to the results-section.

12) Related to the previous remark, the sum of $E_{\text{HeN-In}^}$ and $E_{\text{kin,He}}$ (I assume this is the total kinetic energy of all helium making up the droplet) is well above 0.3 eV. After 1 ps this sum has reduced to well below 0.3 eV. What has happened to the missing energy of approximately 0.05 eV?*

We agree that the sum of $E_{\text{HeN-In}^*}$ and $E_{\text{kin, He}}$ slightly decreases, as is also clearly reproduced by the simulations. This is due to other energy terms of minor contribution in the dynamics simulation that are not considered in figure 4c. They stem mostly from He-He correlation energies, that arise due to the He density waves that are produced following the bubble expansion.

We added the sentence "The minute decrease of the sum of $E_{\text{HeN-In}^*}$ and $E_{\text{kin, He}}$ over time represents energy transferred to He-He interactions (correlation energies)." to the discussion-section in order to account for this unclarity.

Response Reviewer 2

1) *The most significant feature is the expansion of the Helium droplet from 4.5 to 8 Angstrom in response to the photoexcitation of In. On page 4 the main reason for this is stated to be the expansion of the atomic valence electronic wavefunction. To back up this conclusion, the respective valence orbitals participating in the excitation should be shown.*

We agree that without an extended electron density in the excited state, the expansion of the solvation shell would not take place. We calculated the outer electron densities for the full ground and first excited state of indium (see figure 1 below), which clearly show an extended radial electron density (averaged over full solid angles) for the excited $5s^26s$ state, compared to the $5s^25p$. We note however, that this isolated atomic picture is only a zeroth order approximation in comparison to the helium-solvation case, as no dopant-helium interaction is considered. A more sophisticated description is the In-He pairwise interaction energy (see supplementary figure 3), in which the repulsive part at lower distances in the excited state $5s^26s$ hints at a bubble expansion due to Pauli repulsion. However, even this description lacks correlated He-He interactions, that are fully accounted for in the TDDFT simulation that we present in the main manuscript. Because of this we would rather prefer to not show electron densities of the isolated atom in the paper.

2) *On page 6, it would be helpful to the equations how the interaction energies are calculated. Some general remarks on how the dynamics is realized in the program should be given.*

As we are not the developers of the code, we prefer to refer the reader to references [20] and [21], where all simulations, including the calculation of the interaction energies, are described in great detail.

To give some more insight, we extended the description of the HeDFT in the methods-section with a few comments: We mention the important Orsay-Trento functional (accounting for He-He interactions) and added the phrase "by solving the TDDFT equations for the helium and Newton's equations of motion for the impurity" to the description. We further added the formula for calculating the PE peak shift induced by the surrounding He:

$$\text{PE shift (t)} = \int \rho_{\text{He}}(\mathbf{r}, t) E_{\text{In}^*-\text{He}}(\mathbf{r} - \mathbf{r}_{\text{In}^*}(t)) \, d\mathbf{r} - \int \rho_{\text{He}}(\mathbf{r}, t) E_{\text{In}^+-\text{He}}(\mathbf{r} - \mathbf{r}_{\text{In}^+}(t)) \, d\mathbf{r}$$

3) *Authors should give some ideas on how the expansion of the He droplet can help to characterize more complex processes as advertised in the introductory section on page 2. This would certainly justify the broader relevance of the paper.*

To give a prospect, we have extended the conclusion section with the following sentences: "The expansion of the dopant solvation shell will be superimposed on any molecular relaxation dynamics on femtosecond time scales inside the droplet. When applying photoelectron detection, which seems to be a promising observable for intrinsic molecular dynamics inside helium droplets, the photoelectron transients induced by solvation shell dynamics have to be known. The duration of dopant ejection, on the other hand, limits the time frame for which ultrafast reactions inside the quantum fluid can be observed."

Figure 1: The upper panel shows the radial electron densities for the indium ground state ($5s^2 5p$) and the excited state ($5s^2 6s$). Core electrons are considered with an effective core potential (see supporting information), for which reason the electron density plotted here approaches zero at small distances. The lower panel shows the total electron density difference (scaled with r^2). Inner electron density is fully transferred to outer regions in the excited state.

Response Reviewer 3

1) The organization of the manuscript is a bit strange. The dopant is not identified as In until the bottom of page 3. At no point in the manuscript is the choice of In as the dopant discussed or justified. Pump-probe photon energies should be indicated in the text and/or figure caption. There is no "discussion" section; everything is lumped into "conclusions". Just when one thinks the paper is done, there is a paragraph at the end of the conclusions about oscillations of the He bubble. This is the first time in the paper that bubble oscillations are mentioned- where is all this coming from?

We now mention indium as our dopant of choice in the abstract as well as in the introduction of our paper. Pump and probe photon energies are now, in addition to the respective wavelengths, included in the "Methods" section. We agree that the "Conclusion" section consists to a large fraction of discussion and have therefore changed its title to "Discussion", which is also in line with Nature Communications' manuscript guidelines.

Regarding the bubble oscillation, we would like to mention that the oscillation, as well as its period, are announced in the abstract, as well as in the introductory section of the paper.

In order to justify the choice of In as dopant we have added the following sentence to the Methods section: "Indium was chosen as dopant because of its simple electronic structure with one valence electron and because its excited state is symmetric, simplifying the TDDFT simulations, as well as the interpretation."

2) In Fig. 2, the total photoelectron intensity drops significantly after about 200 fs. Why does this occur? The In excited state doesn't disappear, so why should the ionization signal drop?

The drop in intensity can be explained mainly with the cross-correlation between pump and probe pulses, which increases the signal significantly due to the higher intensity. In addition, the PE signal drops in magnitude also after 200 fs, where only a minor influence of the cross correlation is expected. This implies that the ionization probability for larger bubbles is reduced. This however remains a speculation at the moment and will be addressed in future studies. The interpretation of the presented results is, however, not affected as it is based on the PE peak position, which is correctly determined.

We added the sentence "Additionally the total PE signal decreases during the expansion, hinting at reduced ionization probabilities for larger bubbles." to the caption of figure 2.

3) The authors need to do a better job explaining the connection between the long-time PE spectrum and the expected PE energy of 0.30 eV for a gas phase In atom. I assume that this value is the sum of the two photon energies minus the IP of atomic In. However, if there is significant relaxation on the excited state of In after the first photon is absorbed, as implied by Fig. 1 and 4b, then it is not obvious why the PE energy at long times is reduced by this relaxation, in which case it should not match the gas phase value. This is an important point; in the absence of such an explanation, one questions the overall interpretation of the data.

The PE energy at long time delays (0.30 eV) is obtained as the energy difference between the probe

photon energy and the binding energy of the bare atom excited 6s state:

$$E_{e,\text{kin}} = E_{\text{hv,probe}} - (IP_{\text{free}} - E_{6s,\text{free}}),$$

assuming that after dopant ejection the atoms remain in the excited state. The energy relaxation takes place solely inside the He-droplet and is related to "molecular" structure of the In-He_N system. Energy converted during the relaxation from electronic potential into kinetic energy of the He is provided as excess energy of the pump photon with respect to gas phase excitation and therefore the pump photon energy is not included in the energy balance above.

We think that this basic idea of photoelectron spectroscopy is sufficiently explained by the following sentences, which are included in the "TRPES" section: "As depicted in figure 1a, after photoexcitation by a pump pulse the evolution of the excited state is probed by time-delayed photoionization and the PE kinetic energy (red arrows) is measured. ... Because the energies of the electronic states depend on the structure of the He environment, the transient response of the quantum solvent can be sensed with TRPES (see figure 1a)."

Additionally, we added the sentence "PE energies are calibrated with the free atom line, which position is retrieved by subtraction of the excited state binding energy [2] from the probe photon energy." to the methods-section.

4) In the simulations shown in Fig. 3, it is assumed (p.6) that the photoexcited In atom lies 20 Å from the droplet center. What physical reason is there for such a significant displacement? One obvious concern here is that if the droplets comprise 4000 He atoms, they are relatively small, and a 20 Å displacement would put the He atom close to the droplet surface.

As is stated in the text, the 20 Å displacement is justified by a good agreement with the experiment, as deeper dopant locations give rise to much longer, or even absent ejection dynamics in the simulation. Also, 20 Å represents the maximum of the spatial probability distribution in the ground state holding-potential. Position distributions and dependences of the dynamics on position and droplet size are beyond the scope of this manuscript and will be addressed in future studies. We also like to add, that the droplets consisting of 4000 atoms have a radius of about 37 Å, setting the impurity at 20 Å with a maximum bubble radius of about 8 Å therefore results in 9 Å distance to the edge of the droplet.

We have extended our explanation in the methods-section to "For the bubble oscillation dynamics the starting position was chosen to be at 20 Å distance to the centre, which is, based on the In-He_N ground state holding potential, a representative dopant location, and leads to a similar ejection behavior as in the experiment. Both the bubble expansion and the oscillation period are purely local effects and are found to not strongly depend on the dopant location inside the droplet." Further we have added snapshots of supplementary video 2 to the supporting online material, where it can be seen that a starting location of 20 Å still represents a fully immersed dopant.

Reviewer #1 (Remarks to the Author):

Following my previous comments regarding the helium bubble dynamics induced by the excitation of the indium, the authors state in their rebuttal that the compression of the bubble is largely independent of the position of the indium atom. They argue that for the case shown the dynamics is superimposed on the ejection dynamics. Furthermore, they claim that multiple oscillations can be observed when the indium atom is located at the center of the droplet. The authors should show proof for these statements by providing the following information in the SI so that readers can judge for themselves.

1. Movie of the droplet dynamics with the indium atom located at the center up to times that multiple oscillations can be seen.
2. A figure like 4a, showing the droplet radius as function of time revealing multiple oscillations. This should be done for the indium at the droplet center and at the 20 Å off-center position.
3. This figure should also include the calculated PE spectrum analogous to Fig. 3, for both locations, to allow a direct comparison.

Another issue raised in my review of the original manuscript was related to the linewidth of the transition in the photoelectron spectrum. The authors have now provided more details about the experimental setup. The mention that the instrumental linewidth of the spectrometer is 10% of the kinetic energy, which is very large for such a device and is also substantially larger than the value of 4% they have reported before for this instrument. How can this be?

Based on the instrumental width of the spectrometer and the bandwidth of the laser pulses, they calculate an overall width of 0.039 meV for the short time delays. With the information provided on the laser bandwidth I calculate a width of less than 0.032 for long time delays. Inspection of Fig. 2b reveals that the width of the transition is much larger than 0.04 meV. From the graph I estimate it to be double this, 0.08 meV. Clearly, the helium affects significantly the photoelectron energy. Since the instrumental width cannot explain the total width the question remains, how can the PE signal extend to energies that far below the gas phase limit?

The interpretation of the recorded photoelectron is fully based on the accompanying calculations. Whereas these types of calculations have been shown to be able to model dynamics in helium droplets, they have also failed to do so in certain cases. Therefore, all available experimental data should be used to unravel the dynamics. In this case the authors do not use the information encoded in the linewidth of the PE peaks. This information would allow them to unambiguously unravel the contribution of the desorption process to the recorded PE spectra. I agree with the authors that the linewidth cannot be modeled and compared to theory, but having this information in addition to the PE peak positions removes any doubt about the interpretation of the data. The authors should therefore provide this data, if not in the manuscript itself at least in the supporting information.

The dynamics at short time scales are characterized by broad peaks in the PE spectrum whose intensity decays with delay time. While the data up to 150 fs can be affected by cross correlations, in view of the used laser pulses this is not possible for longer times. The authors speculate without giving any justification that this could be due to the ionization probability of larger bubbles being smaller. In my opinion they should also consider the option, for which evidence exist, that slow electrons arising from large bubbles at long delays, have a smaller escape probability from the droplets than fast electrons originating small bubbles at short times.

The authors attribute the large linewidth in the PE spectrum at times between 150 and 500 fs to the fact that not all bubble expansion "trajectories" of the observed ensemble are started at the same time. They should provide an estimate of the broadening caused by this effect and mention the explanation explicitly in text of the manuscript.

In their simulations the authors placed the indium atom 20 Å off-center with the argument that

this position yields the correct time scale for the ejection dynamics and that this corresponds to the maximum of the spatial probability distribution. The timescale for ejection is not known from the experiment and thus it cannot be argued that this position yields the correct timescale. The lowering of the PE energy at long times is compatible with this assumption, but this is only validated by comparing to these same calculations. Therefore additional experimental information is required. As already argued above the PE linewidth as function of delay time could provide this information and the authors should provide this.

In addition, the authors should provide in the SI the calculated spatial probability distribution, and explicitly mention in the text that the choice of 20 A is based on this calculation.

As a minor point, Fig. 1 in the manuscript has not been replaced by a new figure as indicated in the rebuttal.

Reviewer #2 (Remarks to the Author):

The authors addressed all my questions and I recommend publication of the paper. Nevertheless, I would recommend to either include supplementary figure 3 in the main manuscript or to stonger relate to it in the main manuscript. This figure illustrates the origin of the He droplet expansion, while figure 4b in the main manuscript proves that the expansion happens.

Reviewer #3 (Remarks to the Author):

The authors have adequately addressed my concerns and the paper is now suitable for publication.

Response Reviewer 1

We thank reviewer 1 for his detailed comments, which motivated us to show additional results and to present more experimental information in the paper. This includes an analysis of the linewidth and -shape and their temporal change, as well as the transient ion yield, which is a complementary observable for the ejection dynamics. We also show now additional simulations, in particular for photoexcitation at different positions within the droplet. At the same time we want to point out that this paper represents the first femtosecond investigation of dopants inside helium droplets and therefore brings up new questions, such as the ground state spatial distribution of the dopants and their translational dynamics before photoexcitation. These are subject to our current experiments. We are convinced, that the presented dynamics, namely the bubble expansion and oscillation, are sufficiently well described with the presented results and discussion.

1) Following my previous comments regarding the helium bubble dynamics induced by the excitation of the indium, the authors state in their rebuttal that the compression of the bubble is largely independent of the position of the indium atom. They argue that for the case shown the dynamics is superimposed on the ejection dynamics. Furthermore, they claim that multiple oscillations can be observed when the indium atom is located at the center of the droplet. The authors should show proof for these statements by proving the following information in the SI so that readers can judge for themselves.

- 1. Movie of the droplet dynamics with the indium atom located at the center up to times that multiple oscillations can be seen.*
- 2. A Figure like 4a, showing the droplet radius as function of time revealing multiple oscillations. This should be done for the indium at the droplet center and at the 20 Å off-center position.*
- 3. This Figure should also include the calculated PE spectrum analogous to Fig. 3, for both locations, to allow a direct comparison.*

We added Supplementary Movie 3, showing the dynamics of a centre-located In atom, and corresponding text (printed in blue color) to the Supplementary information. Furthermore, Supplementary Note 4 was also included, where we show in Supplementary Figure 6a the solvation shell radius as function of time for both starting locations. In Supplementary Figure 6b calculated PE peak shifts over time are shown for photoexcitation at different distances to the droplet centre. The movie and the two Figures clearly show that multiple oscillations are triggered by photoexcitation in the droplet centre, whereas only one oscillation is observed for a simulated excitation at 20 Å, superimposed on ejection represented by a decrease of the PE peak energy to the bare-atom value. We would like to point out, as mentioned in the supporting information, that the In atom appears to be at rest in the droplet centre because the simulation is based on zero initial velocity. In reality, a velocity of a few meters per second leads to translation to the droplet surface. This velocity is, however, so small that the corresponding distance after the observed times is negligibly small.

In the main text we added to the discussion section: "Photoexcitation of the In dopant in the centre of the droplet induces multiple oscillations and no ejection within the simulated time span (see Supplementary Movie 3 and Supplementary Note 4)."

2) Another issue raised in my review of the original manuscript was related to the linewidth of the transition in the photoelectron spectrum. The authors have now provided more details about the experimental setup. The mention that the instrumental linewidth of the spectrometer is 10% of the

kinetic energy, which is very large for such a device and is also substantially larger than the value of 4% they have reported before for this instrument. How can this be?

The energy resolution of 4% represents the reference value cited in literature for a magnetic bottle spectrometer of the same flight tube length (see M. Mucke et al., Rev. Sci. Instrum. 83, 063106 (2012); doi: 10.1063/1.4729256). A value of $E/\Delta E = 30$ is obtained in the paper for a wider range of electron kinetic energies (c.f., Figure 5 in RSI 83, 063106), corresponding to a relative energy resolution of 3.3%. The characterization measurements also show, however, that this value can only be obtained by a very small laser (synchrotron beam) diameter placed at close distance (1.1 mm) to the permanent magnet and that larger distances lead to both a shift and broadening of the PE spectrum (c.f., Figure 3 in RSI 83, 063106). For our current work we have chosen a larger laser diameter of 1 mm in order to increase the electron signal. Additionally the probe photon energy used in the experiment results in a low kinetic energy of the photoelectrons (~ 300 meV). We ascribe the reduced energy resolution of 10% to these two reasons, it is, however, sufficient to resolve the transient shift of the single In PE peak. Evaluation of the linewidth obtained with bare In atoms (c.f., Supplementary Figure 7b) yields an energy resolution of about 10%, as stated in the methods section.

In the following we combine our response to comments 3 and 5:

3) Based on the instrumental width of the spectrometer and the bandwidth of the laser pulses, they calculate an overall width of 0.039 meV for the short time delays. With the information provided on the laser bandwidth I calculate a width of less than 0.032 for long time delays. Inspection of Fig. 2b reveals that the width of the transition is much larger than 0.04 meV. From the graph I estimate it to be double this, 0.08 meV. Clearly, the helium affects significantly the photoelectron energy. Since the instrumental width cannot explain the total width the question remains, how can the PE signal extend to energies that far below the gas phase limit? The interpretation of the recorded photoelectron is fully based on the accompanying calculations. Whereas these types of calculations have been shown to be able to model dynamics in helium droplets, they have also failed to do so in certain cases. Therefore, all available experimental data should be used to unravel the dynamics. In this case the authors do not use the information encoded in the linewidth of the PE peaks. This information would allow them to unambiguously unravel the contribution of the desorption process to the recorded PE spectra. I agree with the authors that the linewidth cannot be modeled and compared to theory, but having this information in addition to the PE peak positions removes any doubt about the interpretation of the data. The authors should therefore provide this data, if not in the manuscript itself at least in the supporting information.

5) The authors attribute the large linewidth in the PE spectrum at times between 150 and 500 fsec to the fact that not all bubble expansion "trajectories" of the observed ensemble are started at the same time. They should provide an estimate of the broadening caused by this effect and mention the explanation explicitly in text of the manuscript.

With the following text, which we added to the results-section of the main manuscript, we discuss the linewidth of the PE peak within the first picosecond: "The linewidth of the PE spectra is significantly increased and changes within the first picosecond (Figure 2b), which we ascribe to the following four reasons: First, during pump-probe cross correlation of 150 fs, saturation effects and the spectral width of the pump pulse are expected to contribute to the PE linewidth. Second, within the first 500 fs, a peak shift with a maximum slope of about 1 meV/fs in combination with the 150 fs pump-probe cross correlation leads to an expected contribution of about 150 meV. Third, ionization inside the droplet increases the linewidth, given by the Franck-Condon overlap

of the excited and the ionic state (c.f., Figure 1), which seems to be the dominant contribution to the linewidth after 500 fs. Fourth, relaxation of the photoelectrons due to binary collisions with individual He atoms on the way out of the droplet leads to an asymmetric shape of the PE peaks [17]. These decelerated electrons can be seen as wing extending to PE energies below the gas phase value (red line in figure 2b and Supplementary Figure 7b).

Concerning the PE linewidth above 1 ps delay, we added a graph showing the linewidth of the PE peak as function of pump-probe time delay to Figure 3 of the main text. Additionally, we show detailed measurements of the PE peak obtained from the In-He_N system at 0.8 ps and 200 ps and compare them to the PE peak of bare In atoms, which are shown in Supplementary Figure 7b. As discussed in point 6 above, we now show the transient ion yield in Supplementary Figure 7a, an observable providing direct insight into the desorption process.

We added the following text to the main manuscript:

Results-section: "The PE peak width shows a very similar trend (Figure 3, red diamonds) with a steady decrease over time to about 35 meV at long time delays and a temporary increase. Detailed scans of PE peaks at short and long time delays are shown in Supplementary Figure 7b."

Discussion-section: "Dopant ejection is further confirmed by observing a rise in photoion yield on the same time scale (Supplementary Figure 7a) and a transient change of the linewidth of the PE peak (see Figure 3 and Supplementary Figure 7b). "

4) The dynamics at short time scales are characterized by broad peaks in the PE spectrum whose intensity decays with delay time. While the data up to 150 fsec can be affected by cross correlations, in view of the used laser pulses this is not possible for longer times. The authors speculate without giving any justification that this could be due to the ionization probability of larger bubbles being smaller. In my opinion they should also consider the option, for which evidence exist, that slow electrons arising from large bubbles at long delays, have a smaller escape probability from the droplets than fast electrons originating small bubbles at short times.

We have changed the corresponding sentence from

"Additionally, the total PE signal decreases during the expansion, hinting at reduced ionization probabilities for larger bubbles."

to

"Additionally, the total PE signal decreases during the expansion, which might be due to a decreased ionization probability for larger bubbles and/or lower escape probability of slow electrons from larger bubbles at long delays compared to fast electrons from small bubbles at short delays [17,20]."

6) In their simulations the authors placed the indium atom 20 Å off-centre with the argument that this position yields the correct time scale for the ejection dynamics and that this corresponds to the maximum of the spatial probability distribution. The timescale for ejection is not known from the experiment and thus it cannot be argued that this position yields the correct timescale. The lowering of the PE energy at long times is compatible with this assumption, but this is only validated by comparing to these same calculations. Therefore additional experimental information is required. As already argued above the PE linewidth as function of delay time could provide this information and the authors should provide this. In addition, the authors should provide in the SI the calculated spatial probability distribution, and explicitly mention in the text that the choice of 20 Å is based on this calculation.

We provide additional experimental information by showing the transient change of the linewidth

(Figure 3 of the main text), as well as the transient ion signal (Supplementary Note 5 and Supplementary Figure 7a). We consider the ion yield rise over time as most direct observation of the spatial distribution of In atoms within the droplet and the corresponding distribution of ejection times of the observed ensemble. The ion signal rise shows approximately the same temporal change as the lowering of the PE linewidth and the lowering of the PE energy.

Concerning justification of the 20 Å position, we have gained important new insight since the last review. Apparently, the ground state holding potential of the Indium atom inside the droplet depends very sensitively on both minute changes of the In-He pair and even the DFT code used for calculating the holding potential. For comparison, Figure 1 below shows two ground state holding potentials computed with our in-house DFT code using two different basis sets (aug-cc-pV5Z and Def2-QZ), as well as one obtained with the publicly-available BCN-TLS-He-DFT computation package (M. Pi et al., <https://github.com/bcntls2016>, and Ancilotto et al., *Int. Rev. Phys. Chem.* 36, 621–707) for the In-He₄₀₀₀ system. A difference in the dissociation energies of the In-He pair potentials of only 0.4 cm⁻¹ due to different basis sets (aug-cc-pV5Z and Def2-QZ) completely changes the shape of the In-He₄₀₀₀ potential. (This is the topic of a current experimental and theoretical investigation). As the three potentials would result in completely different spatial probabilities, we prefer to not show any of them in the paper. Nevertheless, 20 Å is still the position of choice as is justified by a comparison of calculated PE transients for different starting positions, which are shown in Supplementary Figure 6b (see point 1 above). The 20 Å transient shows the best agreement with our measurement (Figure 3 of the main text) as it shows a single bubble contraction, compared to for example none in the 16 Å transient and two in the 24 Å transient. We note that the strong dependence of the ground state holding potential is caused by the similarity of the In-He and the He-He pair potentials. For the In*-He_N system in its excited state the situation is different and the potential energy surfaces, which are the basis for the presented TD-DFT simulations, are more reliable.

Figure 1: In-He₄₀₀₀ potential energy surfaces as function of distance r of the In atom to the He centre of mass. The potentials are obtained with a helium density functional theory (He-DFT) code based on the Orsay-Trento-density functional. The In-He pair potentials are aug-cc-pV5Z (solid line) and Def2-QZ (dashed line). For comparison, the ground state potential obtained with the BCN-TLS-He-DFT computation package using the aug-cc-pV basis basis set is shown as dash-dotted line.

We added the following text to the results section: "The choice of this position is justified by comparing simulated PE peak transients with different starting positions (see Supplementary Note

4 and Supplementary Figure 6).”

7) As a minor point, Fig. 1 in the manuscript has not been replaced by a new Figure as indicated in the rebuttal.

We apologize for this mistake and have now added the appropriate Figure 1 with the suggested changes to the manuscript.

Response Reviewer 2

The authors addressed all my questions and I recommend publication of the paper. Nevertheless, I would recommend to either include Supplementary Figure 3 in the main manuscript or to stonger relate to it in the main manuscript. This Figure illustrates the origin of the He droplet expansion, while Figure 4b in the main manuscript proves that the expansion happens.

We changed the corresponding paragraph in the discussion-section from

"Photoexcitation increases the radial expansion of the valence electron wave function, thereby pushing the surrounding He away (see Supplementary Movie 1)."

to

"Photoexcitation increases the radial expansion of the valence electron wave function, as is suggested by the strong repulsive part of the In-He pair potential at short distances in the excited state (Supplementary Figure 3). Pauli repulsion between the extended electron density and the closed-shell He thereby pushes the surrounding solvation shell away (see Supplementary Movie 1)."

Reviewer #1 (Remarks to the Author):

The authors have adequately addressed my concerns and I recommend publication of the manuscript.